# Role of Oncogenic Pathways on the Cancer Immunosuppressive Microenvironment and Its Clinical Implications in Hepatocellular Carcinoma

**DOI:** 10.3390/cancers13153666

**Published:** 2021-07-21

**Authors:** Naoshi Nishida

**Affiliations:** Department of Gastroenterology and Hepatology, Kindai University Faculty of Medicine, 377-2 Ohno-Higashi, Osaka-Sayama 589-8511, Japan; naoshi@med.kindai.ac.jp; Tel.: +81-72-366-0221

**Keywords:** cancer, hepatocellular carcinoma, immune evasion, immunotherapy, immune checkpoint inhibitors, oncogenic signaling pathway, molecular targeted agents, genome, epigenome, tumor immune microenvironment

## Abstract

**Simple Summary:**

Hepatocellular carcinoma is known to become resistant to treatments easily by mutations in genes involved in the key cellular pathways targeted by current molecular targeted agents (MTAs). However, the immune checkpoint inhibitor (ICI) is a promising modality for cancer treatment, in which the cancer cells are made recognizable by the immune system. Blockade of the PD1/PDL1 proteins, which help cancers evade the immune system, is currently being tested in clinical trials in combination with MTAs. In this review, several cellular signaling pathways that can alter the immune processes within the tumor and can subsequently affect the patient’s response to ICIs are detailed. This review may help scientists and clinicians to better understand the molecular factors that can influence ICI-based therapy and will help in identifying suitable cases for this type of treatment.

**Abstract:**

The tumor immune microenvironment, including hepatocellular carcinoma (HCC), is complex, consisting of crosstalk among tumor components such as the cancer cells, stromal cells and immune cells. It is conceivable that phenotypic changes in cancer cells by genetic and epigenetic alterations affect the cancer–stroma interaction and anti-cancer immunity through the expression of immune checkpoint molecules, growth factors, cytokines, chemokines and metabolites that may act on the immune system in tumors. Therefore, predicting the outcome of ICI therapy requires a thorough understanding of the oncogenic signaling pathways in cancer and how they affect tumor immune evasion. In this review, we have detailed how oncogenic signaling pathways can play a role in altering the condition of the cellular components of the tumor immune microenvironment such as tumor-associated macrophages, regulatory T cells and myeloid-derived suppressor cells. The RAS/MAPK, PI3K/Akt, Wnt/β-catenin and JAK/STAT pathways have all been implicated in anti-tumor immunity. We also found that factors that reflect the immune microenvironment of the tumor, including the status of oncogenic pathways such as the volume of tumor-infiltrating T cells, expression of the immune checkpoint protein PD-1 and its ligand PD-L1, and activation of the Wnt/β-catenin signaling pathway, predict a response to ICI therapy in HCC cases.

## 1. Introduction

Hepatocellular carcinoma (HCC) is highly refractory and is the third leading cause of cancer-related deaths worldwide [1]. Recent advancements in molecular targeted agents (MTAs) for HCC have dramatically improved the prognosis for patients with this disease. Following the approval of sorafenib as the first MTA for advanced HCC, lenvatinib has also been applied as a first-line systemic chemotherapeutic for HCC, while regorafenib, cabozantinib and ramucirumab have been approved as second-line agents [2]. Because MTAs primarily target molecules involved in oncogenic signaling pathways that play an important role in the development of cancer cells, the development of clones resistant to MTAs can happen easily by genetic mutations and modifications in the specific molecular pathways [3,4]. Hence, additional chemotherapeutic agents would be required.

In contrast, immune checkpoint inhibitors (ICIs) play a role in tumor regression by a different mechanism from that of MTAs [5]. They are known to interfere with the immunosuppressive mechanism to enhance the anti-tumor immune response [6]. Because the target molecules of ICIs are primarily expressed in the stromal cells as well as the cancer cells, ICIs can be effective even for patients who fail to respond well to MTAs or acquire resistance to them, potentially enabling ICIs to complement treatment with MTAs [5,7,8,9]. Although the clinical trial of anti-programmed cell death-1 (PD-1) monotherapy failed to show a significant difference in the survival of patients with advanced HCC compared with conventional MTAs, synergic effects of the combination of different kinds of agents can be expected in several ongoing clinical trials of ICI-based therapy. Based on a successful Phase III clinical trial, the combination of the ICI atezolizumab (an anti-PD-1 antibody) with MTA bevacizumab (an anti-VEGF-A antibody) was approved as a first-line therapy for unresectable HCC [10,11].

Because of the complexity of cancer immunity, where immune cells, tumor cells and other types of stromal cells affect each other, understanding the immune microenvironment of the tumor is difficult [5]. While it has been considered that oncogenic mutations in tumor cells do not directly affect the outcome of ICI therapy, recent reports have suggested that mutation-induced changes in the tumor phenotype can affect the tumor–stroma interactions through alterations in the expression of immunosuppressive cytokines, chemokines, receptors and metabolites, thereby potentially affecting the tumor immune microenvironment [5]. Thus, the anti-tumor effect of MTAs in combination therapy with ICIs can be attributed to the direct action of MTAs on the HCC cells, as well as the reduction in the immunosuppressive nature of the tumor microenvironment through the inhibition of specific oncogenic signals [12].

To understand the significance of oncogenic signaling in the establishment of an immunosuppressive tumor microenvironment, and for the application of this knowledge to the treatment of HCC, this review focused on the role of specific genetic mutations involved in the oncogenic pathways responsible for anti-tumor immunity, and the current status of and perspectives on the combination of ICIs and MTAs for the treatment of HCC.

## 2. Cellular Components and Molecules Associated with an Inhibitory Tumor Immune Microenvironment

Oncogenic signals affect the expression of several immune-related molecules, including immune regulatory receptors, ligands, growth factors and other humoral factors, which affect diverse stromal cells as well as cancer cells. The cellular components of tumors and their states are major players in the regulation of the tumor immune microenvironment. Therefore, to better understand the impact of oncogenic signals on anti-cancer immunity, the functions of the stromal cells involved in the immune microenvironment of tumors are briefly discussed here.

### 2.1. Regulatory T-Cells

Regulatory T-cells (Tregs) are CD4^+^ T-cells characterized by the expression of the transcription factor Foxp3. They can be induced in tumor tissues through growth factors and cytokines, such as transforming growth factor β (TGF-β), interleukin 10 (IL-10) and vascular endothelial growth factor (VEGF), and inhibit immune responses through various mechanisms [13]. In particular, Tregs express the inhibitory immune checkpoint molecule cytotoxic T-lymphocyte (associated) antigen 4 (CTLA-4), which plays a critical role in the regulation of T cell-mediated anti-tumor immunity. Generally, T cell activation occurs through binding of the co-stimulatory factor B7 (CD80/CD86) on antigen-presenting cells and CD28 on T-cells, in addition to T-cell receptor (TCR) recognition of major histocompatibility complex (MHC)-presented antigens. Binding of CD80/CD86 on dendritic cells (DCs) with CTLA-4 on Tregs results in the inhibition of DC maturation. In addition, the membrane molecule CD25 (IL-2 receptor subunit) on Tregs induces the depletion of IL-2 and suppression of cytotoxic T-cells (CTLs) by immunosuppressive cytokines such as TGF-β and IL-10, and cytotoxic secretions such as granzyme B and perforin released by Tregs [13]. A subtype of HCC that showed predominant expression of an mRNA related to Treg response has been reported [14]. Tregs also secrete the epidermal growth factor receptor (EGFR) ligand amphiregulin, which can promote the growth of HCC cells carrying EGFR in an autocrine manner [15]. Tregs also express VEGF receptor 2 on their surface, and the VEGF signal induces the expansion of this type of T cell [16].

### 2.2. Myeloid-Derived Suppresor Cells

Myeloid-derived suppressor cells (MDSCs) are a heterogeneous population of immature myeloid cells that suppress tumor immunity and can be induced by VEGF [16]. Via their increased arginase activity, degradation of arginine, and uptake of tryptophan, cysteine and other amino acids required for T-cell activation, MDSCs reduce the concentrations of these amino acids in the tissue microenvironment, thereby inhibiting the propagation and activation of T cells [17]. In addition, MDSCs produce TGF-β and IL-10, inducing Tregs and inhibiting natural killer (NK) cell function [18]. Furthermore, MDSCs induce the immunosuppressive M2 macrophages by secreting IL-10, which, in turn, downregulates IL-12 production by tumor-associated macrophages (TAMs) [19,20].

### 2.3. Tumor-Associated Macrophages

Generally, two types of macrophages exist in tumor tissues: M1 macrophages and M2 macrophages. Interferon-γ (IFN-γ) and Type 1 helper cell (Th1) cytokines induce the differentiation of inflammatory monocytes into M1 macrophages. Meanwhile, Type 2 helper cell (Th2) cytokines such as IL-4 and IL-13 promote the differentiation of tissue-resident monocytes into M2 macrophages [20]. In tumor immunity, M1 macrophages produce inflammatory cytokines such as tumor necrosis factor α (TNF-α), IL-6 and IL-12, and exert an anti-tumor effect, whereas M2 macrophages produce immunosuppressive cytokines such as IL-10 and TGF-β, and inhibit anti-tumor immune reactions [20]. The microenvironment in cancer is prone to inducing M2 polarization, which is a characteristic phenotype called tumor-associated macrophages (TAMs). The crosstalk between MDSCs and TAMs induces high IL-10 and low IL-12 levels. In addition, naïve CD4^+^ T cells differentiate into Th2 cells that can produce IL-4 [21]. These processes result in the development of M2 macrophages, which is a disadvantageous state for tumor immunity. A high level of IL-10 induces the downregulation of human leukocyte antigen (HLA) Class II antigens and reduces the antigen presentation capacity of DCs [21]. It also expands the Treg population and inactivates natural killer (NK) cells. Additionally, the TGF-β secreted by MDSCs induces the expression of the inhibitory receptor T-cell immunoglobulin and mucin domain 3 (TIM-3) on TAMs [20].

### 2.4. Cancer-Associated Fibroblasts and Vascular Endothelial Cells

Cancer-associated fibroblasts (CAFs) have proangiogenic activity through the production of extracellular matrix and matrix metalloproteinases; they play a role in tissue remodeling [22]. They also inhibit NK cell function through the production of prostaglandin E2 (PGE2) and indoleamine-2,3-dioxygenase (IDO) [23]. Indoleamine-2,3-dioxygenase is an enzyme involved in tryptophan metabolism, and a reduced level of tryptophan in tumors inhibits local T-cell activation. Hence, crosstalk between CAFs and TAMs also plays a role in immunosuppression. CAFs produce IL-8 and cyclooxygenase-2 (COX2), which lead to the release of TNF and platelet-derived growth factor (PDGF) from TAMs, and further activation of CAFs.

### 2.5. Other Stromal Cells

The vascular endothelium is stimulated by angiogenic growth factors such as VEGF and PDGF. It stimulates Tregs and MDSCs in tumor tissues via the production of TGF-β, VEGF and the chemokine C-X-C motif chemokine 12 (CXCL12) [20]. Hepatic satellite cells (HSCs), which generally play a critical role in liver fibrogenesis, also participate in the induction of Tregs and MDSCs by releasing hepatocyte growth factors [24,25]. HSCs also produce amphiregulin and CXCL12, which induce Tregs and MDSCs, respectively [26]. A subset of DCs with high expression of CTLA-4 was also observed in HCC tissues, which may carry immune tolerogenic effects through the production of IL-10 and IDO.

### 2.6. Immunosuppressive Metabolites

As shown above, the concentrations of metabolites from cancer cells and stromal cells strongly affect the immune state of the tumor. Cyclic adenosine monophosphate (cAMP), which accumulates in tumor tissues, inhibits CD4^+^ and CD8^+^ T-cell responses and macrophage activation, and enhances the Treg response by binding to adenosine A2A receptors [27]. In addition, due to the hypoxic environment in tumor tissues, cAMP upregulates the enzyme COX-2, which synthesizes PGE2 from arachidonic acid. Subsequently, PGE2 binds to prostaglandin E receptor 4 on T-cells and affects T-cell activation and cytokine production [20].

MDSC- and TAM-derived arginase hydrolyzes arginine in the urea cycle and inhibits the function of CTLs via this deficiency in L-arginine. In tumor tissues, a hypoxic environment results in the expression of hypoxia-inducible factor 1α (HIF-1α), which is known to activate arginase [17]. Additionally, IDO is reported to be produced by DCs, macrophages, CAFs, vascular endothelial cells and HCC cells via inflammatory cytokines [28]. As previously stated, IDO inhibits T-cell activation and amplification via depletion of tryptophan and stimulates the differentiation of naïve CD4^+^ T-cells into Tregs [20,29].

### 2.7. Immune Checkpoint Molecules

Immune checkpoint molecules regulate excessive T-cell activation and help to maintain immune homeostasis. In cancer cells, however, these immune checkpoint molecules help tumors evade the immune response. Many immune checkpoint molecules and their ligands have been identified, as summarized in Figure 1. Of these, CTLA-4, programmed cell death-1 (PD-1) and its ligand, programmed cell death-ligand 1 (PD-L1), are believed to play central roles in tumors’ immune evasion [5,20]. The induction of immune checkpoint molecules is regulated by environmental factors as well as cell signaling. For example, extracellular stimulation of IFN-γ and hypoxia-induced HIF-1 can enhance the expression of PD-L1 in cancer cells, MDSCs and TAMs [30]. The binding of PD-L1 to PD-1 on TAMs induces the release of the immunosuppressive cytokine IL-10. Additionally, activation of the phosphatidylinositol-3 kinase (PI3K)–Akt pathway also reportedly induces PD-L1, and the loss-of-function mutation of phosphatase and tensin homolog deleted from chromosome 10 (PTEN), a regulator of the PI3K–Akt pathway, is associated with the expression of PD-L1 in cancer [31,32]. We have also shown that activating mutations in PI3KCA are associated with PD-L1 expression in HCC cells [33]. TIM-3, lymphocyte-activation gene 3 (LAG-3) and the B- and T-lymphocyte attenuator (BTLA, CD272) are also known as co-inhibitory molecules on activated T-cells, based on their association with galectin-9, MHC Class II and herpesvirus entry mediator (HVME), respectively (Figure 1). These suppressive receptors are observed in tumor-infiltrating lymphocytes (TILs) in HCC tissues and are considered to be markers of exhausted T-cells [33,34,35].

## 3. Unique Aspects of Immunological Characteristics in the Liver and Hepatocarcinogenesis

Although ICIs are becoming one of the key agents for the treatment of HCC, the response to this type of agent is still unsatisfactory in the majority of HCC cases compared with other types of malignancies [36]. The relatively low response rate to the ICIs can probably be attributed to the low antigenicity of HCC, as tumor mutation is not high in this type of tumor [36,37]. In addition, as the liver needs to be immunotolerant to nonpathological and persistent inflammation, it carries tolerance mechanisms to immune reactions, including cancer immunity.

The liver is continually exposed to the pathogen and microbe components from the gut via the blood supply of the portal vein. In this situation, the liver limits hypersensitivity to food-derived antigens and components of the intestinal flora to prevent excessive tissue damage and maintain systemic tolerance [38]. Chronic infection with hepatitis virus and persistent stimulation by metabolites further induce immune suppression in the liver, which is one of the unique characteristics of the underlying condition of hepatocarcinogenesis [39]. Resident macrophages (Kupffer cells) play a key role in hepatic tolerance through the production of anti-inflammatory cytokines, leading to downregulation of co-stimulatory molecules. This immunological environment of the liver results in the development of fully exhausted T-cells, where suppressive anti-tumor immunity is not susceptible to rescue by ICIs [36,39]. It has also been reported that CD8^+^ PD-1^+^ T-cells in NASH livers show a lack of immune surveillance and tissue-damaging function, which contribute to the increase in HCC emergence upon anti-PD-1 treatment in a NASH mouse model [40]. Augmentation of CD8^+^ PD-1^+^ T-cells was also observed in human NASH; a worse outcome in HCC patients treated with anti-PD-1 antibodies was observed [40]. Although the details of the difference in the response to ICIs between virally induced and NASH-induced HCCs are still unknown, it is possible that different amounts and quality of antigens and the difference in the liver microenvironment, such as the balance between partially exhausted and fully exhausted T-cells, may be involved in the outcome on ICIs [39,41,42].

In addition, ICI may not be effective or may even exacerbate the disease in some HCC patients. It is reported that blockade of the PD-1 and PD-L1 interaction may induce an expansion of PD-1^+^ Tregs isolated from the liver of patients with chronic hepatitis C, because PD-1 on Tregs generally plays a role in the regulation of the CD4^+^CD25^+^FoxP3^+^ T-cells [43]. Therefore, a blockade of the binding of the ligand with PD-1 on Tregs may result in further suppression of anti-tumor immunity [44]. More importantly, PD-1^+^ Tregs may be involved in the hyperprogression of tumors in gastric cancer patients treated with anti-PD-1 antibodies [44]. As hyperprogression on anti-PD-1 antibodies has also been reported in HCC cases, ICI can be even detrimental in such cases [45].

## 4. Signaling Pathways and the Immune Microenvironment of Tumors

Alterations in oncogenic signaling in cancer not only trigger abnormal differentiation and cell proliferation, but also play a crucial role in the immune evasion of tumors [12]. Cancer-related signaling affects the state of the tumors’ immune components via cytokine, chemokine and growth factor production. To date, genetic alterations in several signaling pathways observed in cancers have been reported to affect the tumor immune microenvironment.

### 4.1. RAS/MAPK Signaling Pathway

In malignant melanomas, activating mutations in BRAF (BRAF^V600E^) induce constitutive activation of the mitogen-activated protein kinase (MAPK) pathway, which stimulates immune-tolerant DCs and inhibits CD8^+^ T-cells via the expression of the immunosuppressive cytokines IL-6 and IL-10, as well as via VEGF [46]. This effect has been reported to be inhibited by BRAF inhibitors and VEGF inhibitors. Furthermore, RAS/MAPK signaling inhibits antigen presentation on tumor cells, and inhibition of this pathway is associated with the recovery of MHC expression by IFN-γ in malignant melanomas and breast cancer [47,48]. Meanwhile, in a murine model of pancreatic cancer, activating mutations in KRAS (KRAS^G12D^) induced MDSCs via the production of granulocyte macrophage colony-stimulating factor (GM-CSF) and inhibition of CD8^+^ T-cell infiltration into tumor tissues, which contributed to the establishment of an immunosuppressive tumor microenvironment [49]. In fact, GM-CSF is known to be upregulated in human pancreatic intraepithelial neoplasia and pancreatic cancer cells [49]. An association between activating mutations in KRAS and resistance to ICIs has also been reported in colorectal cancer. KRAS-mediated repression of interferon regulatory factor 2 (IRF) results in the high expression of the chemokine C-X-C motif ligand 3 (CXCL3), which induces MDSCs that express C-X-C motif chemokine receptor 2 (CXCR2) as the receptor of CXCL3 in tumor tissues [50]. In this manner, activation of KRAS induces MDSC-mediated resistance to antitumor immunity in patients with colorectal cancer (Figure 2).

### 4.2. PI3K/Akt Signaling Pathway

Activation of the P13K/AKT signaling pathway is involved in critical cellular functions, including survival, inhibition of apoptosis and proliferation. Activation of P13K/AKT is one of the common features of cancer. PTEN, which regulates this pathway, demonstrates loss-of-function mutations in various cancers. In melanomas, mutations in PTEN are related to resistance to anti-PD-1 antibody treatment and are correlated with a reduced volume of CD8^+^ TILs [51]. In a murine model, PI3K-β inhibitors improved sensitivity to treatment with anti-PD-1 antibodies and anti-CTLA4 antibodies when the resistance was induced by loss-of-function mutations in PTEN [51]. While loss of PTEN function is associated with the induction of various immunosuppressive cytokines, it also induces VEGF, which is presumed to be the mechanism by which immunosuppression is induced. In bladder cancer, activating mutations in PI3KCA are associated with a reduction in TIL volume, while PI3K inhibitors lead to an increase in TILs [52]. PI3K inhibitors have also been reported to inhibit Tregs and induce the differentiation of M2 macrophages into the M1 phenotype [53]. In addition, by inducing PD-L1 expression, both the RAS/MAPK and the PI3K/AKT signaling pathways can be involved in the suppression of anti-tumor immunity. We have previously reported that while activating mutations of the PI3K/AKT signaling pathway are associated with increased expression of PD-L1 in HCC, and the volume of TILs is generally high in PD-L1-positive HCC, TILs are deficient in HCCs with activating mutations of the PI3K/AKT signaling pathway [33]. Therefore, aside from external stimulation such as by IFN-γ, induction of PD-L1 is likely to be attributable to genetic mutations in the PI3K/AKT pathway in this setting (Figure 2).

BRAF^V600E^ induces immune-tolerant dendritic cells (DCs) via the induction of IL-6, IL-10 and VEGF, ultimately inhibiting the action of CD8^+^ T-cells. In addition, KRAS^G12D^ induces MDSCs via GM-CSF production, thereby inhibiting CD8^+^ T-cells from infiltrating tumor tissues. In colorectal cancer, KRAS activation is known to induce CXCL3 expression and the induction of MDSCs with CXCR2, the receptor of CXCL3. In contrast, PI3K/AKT signaling activation is associated with VEGF expression and a decrease in tumor-infiltrating lymphocyte (TIL) volume, and has been reported to induce Tregs and inhibit the shift of M2 macrophages into M1 macrophages. In addition, PI3K/AKT signaling induces PD-L1 expression. Under the activation of WNT/β-catenin signaling, activating transcription factor 3 (ATF3)-mediated CCL4 downregulation is considered to reduce the migration of CD103^+^ dendritic cells into the tumor and reduce CD8^+^ TILs in melanoma. CCL5 was suggested to be involved in HCC. The activation of the WNT/β-catenin signaling pathway has also been reported to be involved in the formation of an immune suppressive tumor microenvironment through the upregulation of IL-10. Activation of the transcription factors YAP/TAZ, which are regulated by Hippo signaling, upregulate PD-L1 and are involved in CXCL5-mediated induction of MDSCs. In addition, YAP is involved in the induction of M2 macrophages by enhancing the transcription of CCL2.

### 4.3. Wnt/β-Catenin Signaling Pathway

ICIs are presumed to be insufficiently effective in cases where infiltration of CD8^+^ T-cells in tumors is lacking, and an analysis of human melanomas has revealed that activation of the Wnt/β-catenin signaling pathway is associated with reduced TILs in tumors. In melanomas with activating mutations in the Wnt/β-catenin signaling pathway, C-C chemokine ligand 4 (CCL4) is downregulated, which reduces the migration of CD103^+^ DCs and leads to a deficiency in CD8^+^ TILs. Furthermore, the transcriptional repression of CCL4 was attributed to the activation of activating transcription factor 3 (ATF3) as a result of the activation of the β-catenin signaling pathway [54]. Activating mutations of the Wnt/β-catenin pathway occur frequently in HCC, and immunosuppression in the tumor microenvironment based on activation of this signal in liver cancer is presumed to occur via downregulation of CCL5 [55]. In a murine HCC model, induction of CCL5 increased the number of DCs and CD8^+^ T-cells in tumors (Figure 2). In melanomas, activation of Wnt/β-catenin signaling also reportedly led to the upregulation of IL-10 through the binding of β-catenin/T-cell factor (TCF) on the IL-10 promoter, thereby contributing to the formation of an immunosuppressive environment [56].

Aside from the altered Wnt/β-catenin signaling in tumor cells, activation of this signaling pathway reportedly disturbs the effector function of CD8^+^ T-cells and induces the exhausted T-cell phenotype in HCC and colorectal cancers, which contributes to the establishment of immune suppressive tumor microenvironment [41]. Interestingly, neutralization of a canonical Wnt ligand, Wnt 3a, enhances the T-cell response through the rescue of DC activation, resulting in tumor regression in a mouse model [42].

### 4.4. MYC Gene

The transcription factor c-myc regulates the expression of genes necessary for cell proliferation and survival. In many cancers, amplification and overexpression of c-myc has been observed; these are involved in inducing the expression of immune checkpoint molecules such as PD-L1 and CD47 [57]. CD47 is a cell surface glycoprotein that regulates phagocytosis by binding to signal regulatory protein alpha (SIRP-α), which is specific to macrophages and DCs. Thus, overexpression of c-myc is involved in the immune evasion of cancer cells through CD47 and PD-L1.

### 4.5. Chromatin Remodeling Pathway

Genomic DNA is stored in the nucleus as chromatin. During transcription, replication or repair, alterations of the chromatin structure by chromatin remodeling regulate the access of transcription factors to the DNA. The SWItch/sucrose non-fermentable (SWI/SNF) complex is a chromatin remodeling factor that induces the alteration of nucleosomes via ATP hydrolysis. Genetic abnormalities in SWI/SNF complex subunits are frequently observed in HCC and other human tumors [33], and loss-of-function mutations in the polybromo 1 (PBRM1) gene involved in the SWI/SNF complex are common in renal cancer. Intriguingly, this PBRM1 mutation is associated with the therapeutic effect of ICIs in renal cancer. *PBRM1*-deficient renal cancers show altered transcriptional expression in the JAK/STAT (Janus kinase/signal transducers and activators of transcription) and immune signaling pathways [58].

### 4.6. JAK/STAT Signaling Pathway

The JAK/STAT pathway, which transmits signals that are crucial for growth, differentiation, survival and immunity, is altered in many types of malignancy. The downstream transcription factor, STAT3, acts on the PD-L1 promoter, thereby inducing upregulation of PD-L1 in cancer cells. In melanomas, JAK1 and JAK2 mutations inhibit signals from interferon receptors and reduce antigen presentation on tumor cells, which results in resistance to ICI therapy [59]. Meanwhile, β2-microglobulin gene mutations have been reported to induce resistance to ICI treatment via the loss of MHC Class I antigen expression on the cell surface [60].

### 4.7. Hippo Signaling Pathway

Hippo signaling, which is involved in the regulation of growth and differentiation as well as in controlling organ size, is dysfunctional in many malignancies. Reduced Hippo signaling is also associated with cancer’s immune evasion. Hippo signaling regulates yes-associated protein (YAP) and “transcriptional coactivator with PDZ-binding motif” (TAZ), the activation of which leads to the expression of PD-L1 and stimulates MDSCs carrying CXCR2, by upregulation of its ligand, CXCL5 [61]. In a murine model of HCC, YAP was reported to be associated with tumor immunosuppression via the induction of M2 macrophages resulting from enhanced transcription of CCL2 (Figure 2) [62].

### 4.8. DNA Repair Pathway

It is well known that cancers carrying mutations in DNA mismatch genes induce a large number of neoantigens that are attributed to the emergence of a variety of passenger mutations that occur in the microsatellite sequences of the DNA, where anti-tumor immunity is enhanced. Therefore, microsatellite instability is a biomarker for efficacy in the treatment of ICIs [63]. Similarly, cancers with a high mutation burden (TMB) are also markers of tumors with an active immune microenvironment because of their high antigenicity [64]. Recently, it was reported that loss-of-function mutations in the breast cancer susceptibility (*BRCA*) 1 and *BRCA 2* genes, which are involved in the homologous recombination pathway of DNA repair, are also markers of a high TMB and could be predictors of the outcome of ICI-based treatment [65]. From this perspective, alterations in DNA repair pathways are critical for the establishment of high antigenicity and “immune hot” status in cancer.

### 4.9. VEGF Signaling

In tumors, external stimulation can lead to the production of growth factors. In HCC, tissue hypoxia leads to the production of VEGF via the activation of hypoxia-inducible factor 1 (HIF-1), resulting in tumor angiogenesis. The cellular components of tumors that suppress tumor immune responses, such as MDSCs, Tregs and TAMs, express VEGF receptors; therefore, inhibition of VEGF/VEGFR can alter anti-tumor immunity [11]. Anti-PD-1 antibodies and anti-VEGFR-2 antibodies have been reported to have a synergistic effect in murine models of HCC. Anti-VEGFR-2 antibodies induce an increase in CTLs and a decrease in TAMs and Tregs [66]. Atezolizumab + bevacizumab is expected to combine the effects of ICIs with inhibition of VEGF signaling to alter immunity. Pembrolizumab + lenvatinib, a multikinase inhibitor (MKI) with a powerful antiangiogenic effect, and atezolizumab + cabozantinib, which is capable of blocking angiogenesis through the inhibition of VEGFR and AXL, and camrelizumab (an anti-PD-1 antibody) + apatinib (a selective VEGFR2-tyrosine kinase inhibitor) are in Phase III clinical trials, while avelumab (an anti-PD-L1 antibody) + axitinib (which strongly inhibits VEGFR) are undergoing Phase I/II trials (Table 1). ICIs and agents with an anti-VEGF/VEGFR effect are currently the most promising combination therapies for HCC because of their synergistic effect on cancer immunity [6,11].

## 5. Signaling Pathway Abnormalities and the Immune Microenvironment in HCC

In a mouse model of HCC, it has been shown that activation of Wnt/β-catenin signaling induces reduced migration of CD103^+^ DCs and CD8^+^ TIL deficiency via downregulation of CCL5. Previous reports have also shown that Wnt/β-catenin activation is associated with the reduced expression of T cell-derived genes in HCC tissues. Therefore, HCC with activated Wnt/β-catenin signaling is unlikely to respond to ICIs because of the “immune cold” phenotype [67]. In fact, post-ICI therapy outcomes are reported to be poor in cases of HCC with Wnt/β-catenin activation [68]. Using a cohort of HCC cases from The Cancer Genome Atlas (TCGA), we determined that the expression of T cell-related genes was low in cases of HCC with activating mutations in Wnt/β-catenin (Figure 3). In addition, in an analysis of HCC tissues, we determined that HCCs with activating mutations in Wnt/β-catenin pathway genes are significantly deficient in CD8^+^ TILs [33]. However, we did not find CD8^+^ TILs to be associated with mutations in any other oncogenic signaling pathways (Table 2).

In a transcriptome analysis, we reported that Wnt/β-catenin signaling activation was associated with the decreased expression of gene sets related to T-cell priming/activation, IFN-γ response, immunosuppression and Tregs; it was most significantly associated with the downregulation of genes related to the IFN-γ response in multivariate analysis [69]. These data are consistent with the deficiency in CD8^+^ T-cells in HCC tissues. In addition, we also reported that activating mutations in the Wnt/β-catenin pathway is negatively associated with PD-L1 expression in HCC [33]. As the expression of PD-L1 can be induced by the stimulation of IFN-γ, the lack of PD-L1 expression in HCC with Wnt/β-catenin activation can probably be attributed to the low degree of CD8^+^ TILs that should secrete IFN-γ [69]. On the other hand, a previous study found that mutations in genes involved in chromatin remodeling, such as AT-Rich Interaction Domain 2 (*ARID2*), were also associated with an immunosuppressive tumor microenvironment through the expression of genes involved in the induction of M2 macrophages [14], although there were no associations between mutations of the genes involved in chromatin remodeling and the degree of CD8^+^ TILs as well as PD-L1 expression [33]. As mutations of *ARID2* are reportedly associated with the TAM subclass of HCC, the immune suppressive mechanism in HCCs with *an ARID2* mutation should be different from that of CTNNB1 [14]. In contrast, PD-L1-positive HCCs often have high levels of CD8^+^ TILs [33]. This may be due to the fact that PD-L1 expression in HCC cells can be mainly attributed to stimulation by the IFN-γ from TILs. It is possible that, under continuous immune response to cancer cells, many CD8^+^ TILs are prone to expressing multiple inhibitory receptors (PD1, TIM-3, LAG-3) that result in the exhausted phenotype of T-cells [33]. In many cases, PD-L1 expression is considered to be a favorable prognostic factor of ICI therapy, suggesting that blockade of the PD-1/PD-L1 response could, at least partially, activate the T-cell immune response, even if the immune cells express additional inhibitory receptors. Indeed, we found that the absence of activating mutations in Wnt/β-catenin pathway genes, a high CD8^+^ TIL volume and PD-L1 expression were associated with long progression-free survival of HCC patients on anti-PD-1 antibody therapy, regardless of the expression of other inhibitory receptors, such as TIM-3 and LAG-3 [69]. In this way, assessments of gene alterations in cellular signaling pathways are not only useful for finding suitable MTAs that act on the altered cellular signal, but may also, theoretically, serve to predict the response to ICI therapy, based on the tumor immune microenvironment.

## 6. Conclusions

Alterations of cell signaling pathways play a critical role not only in the development of a malignant phenotype in cancer cells but also in the determination of anti-cancer immunity. In a Phase III clinical trial with HCC patients, ICI monotherapy failed to yield a significant anti-cancer response, suggesting that ICIs will be used primarily in combination therapy [6]. It has been speculated that the “immune cold” phenotype of the tumor microenvironment is critical for poor prognosis with ICIs, where activation of the Wnt/β-catenin signaling pathway plays an important role. From this point of view, understanding the response of HCCs carrying Wnt/β-catenin mutations to combination therapy with ICI and MTA is clinically important but has not been clarified yet. Currently, a combination of atezolizumab + bevacizumab is applicable for unresectable HCC; the efficacy of this combination on HCCs showing the “immune cold” phenotype is now under investigation [36]. In addition, aside from atezolizumab + bevacizumab, many ongoing clinical trials have examined combinations of ICIs and MTAs for HCCs that are refractory upon ICI monotherapy, mainly with MTA showing an anti-angiogenic effect (Table 1). However, future trials are likely to examine combinations of ICIs with agents that inhibit other oncogenic pathways that are critical for hepatocarcinogenesis, such as the Wnt/β-catenin pathway, the RAS/MAPK pathway and the PI3K/AKT pathway.

## Figures and Tables

**Figure 1 cancers-13-03666-f001:**
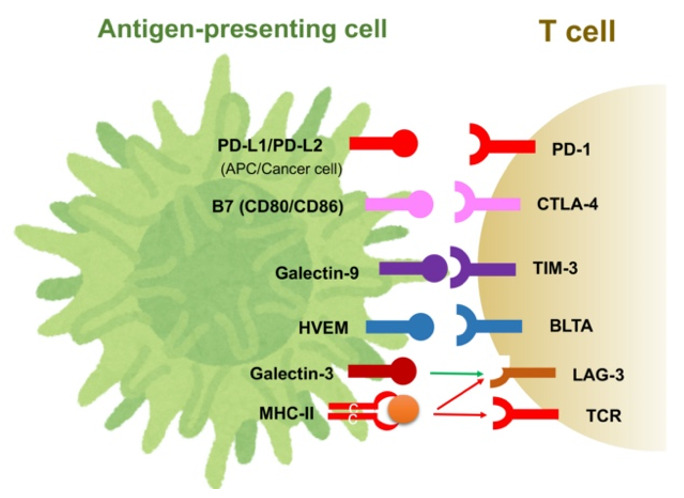
Immune checkpoint molecules and their ligands.

**Figure 2 cancers-13-03666-f002:**
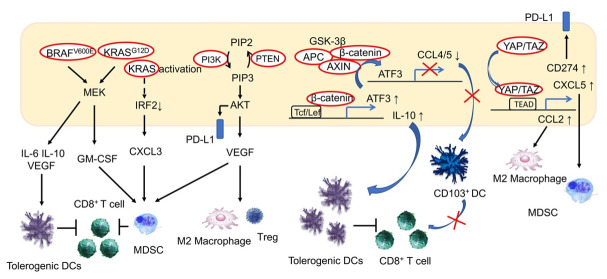
Effects of oncogenic signal activation on the tumor immune environment.

**Figure 3 cancers-13-03666-f003:**
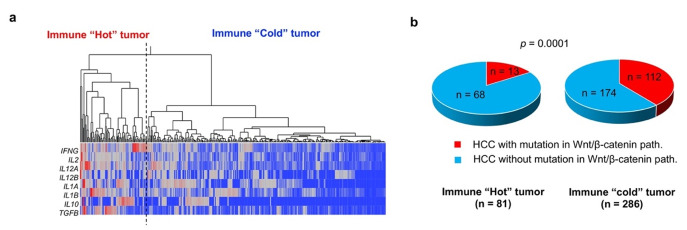
Tumor-infiltrating lymphocyte status and abnormal WNT/β-catenin activation. Hierarchical cluster analysis was used to classify HCCs based on the expression of eight T cell-related cytokine mRNAs obtained from the TCGA transcriptome dataset (RNA-seq V2 RSEM) (**a**) “Immune hot” and “immune cold” refer to HCCs with high and low levels of T cell-related gene expression, respectively. Thus, “immune hot” indicates a large volume of tumor-infiltrating lymphocytes (TILs), while “immune cold” suggests that TILs are deficient. (**b**) The presence or absence of activating mutations in WNT/β-catenin pathway genes are determined based on the presence or absence of *CTNNB1, AXIN1* and *APC* mutations in the TCGA provisional dataset obtained. Activating mutations in the WNT/β-catenin pathway are more frequently detected in “immune cold” HCCs than in “immune hot” tumors (*p* = 0.0001). The TCGA dataset used in the present study was downloaded in September 2019.

**Table 1 cancers-13-03666-t001:** Clinical trials for combinations of molecular targeted agents and immune checkpoint inhibitors in hepatocellular carcinoma.

NCT Number ^1^ (Trial Name)	MTAs/ICIs ^2^	Targets of MTAs	Setting
Phase I/II			
NCT03299946 (CaboNivo)	Cabozantinib/ Nivolumab	TKI for VEGFR2, MET, AXL, etc.	neoadjuvant
NCT03170960 (COSMIC-021)	Cabozantinib/ Atezolizumab	Same as above	First-line
NCT04442581	Cabozantinib/ Pembrolizumab	Same as above	First-line
NCT01658878 (CheckMate 040)	Cabozantinib/Nivolumab±Ipilimumab	Same as above	First -and second-line
NCT03289533 (VEGF Liver 100)	Axitinib/ Avelumab	TKI for VEGFR1-3, PDGFR, c-kit, etc.	First-line and AFP ≥ 400 ng/mL
NCT03841201, NCT03418922	Lenvatinib/ Nivolumab	TKI for VEGFR1-3, FGFR1-4, etc.	First-line
NCT03347292 (Bayer 19497)	Regorafenib/Pembrolizumab	TKI for VEGFR1-3, TIE2, PDGFR, c-kit, RET, etc.	First-line
NCT04310709 (RENOBATE)	Regorafenib/Nivolumab	Same as above	First-line
NCT04183088	Regorafenib/Tislelizumab	Same as above	First-line
NCT03941873	Sitravatinib/Tislelizumab	TKI for VEGFR2, c-kit, AXL, etc.	First-line and later
NCT03475953 (REGOMUNE)	Regorafenib/Avelumab	TKI for VEGFR1-3, TIE2, PDGFR, c-kit, RET, etc.	Second-line
NCT04170556 (GOING)	Regorafenib/Followed by Nivolumab	Same as above	Second-line
NCT03539822 (CAMILLA)	Cabozantinib/Durvalumab	TKI for VEGFR2, MET, AXL, etc.	Second-line
NCT02572687	Ramucirumab/Durvalumab	Ab for VEGFR2	Second-line and AFP ≥ 1.5x ULN
NCT02423343	Galunisertib/Nivolumab	TKI for TGβR1	Second-line and AFP ≥ 200 ng/mL
Phase III			
NCT04102098 (IMbrave050)	Bevacizumab/Atezolizumab	Ab for VEGFA	Adjuvant
NCT03847428 (EMERALD-2)	Bevacizumab/±Durvalumab (vs. placebo)	Same as above	Adjuvant
NCT03713593 (LEAP-002)	Lenvatinib/Pembrolizumab (vs. Lenvatinib)	TKI for VEGFR1-3, FGFR1-4, etc.	First-line
NCT03755791 (COSMIC-312)	Cabozantinib/Atezolizumab (vs.orafenib or. Cabozantinib)	TKI for VEGFR2,MET, AXL, etc.	First-line
NCT03764293	Apatinib/Camrelizumab (vs. sorafenib)	TKI for VEGFR2	First-line

^1^ National Clinical Trial number (ClinicalTrials.gov registry number). ^2^ MTA: molecular targeted agent; ICI: immune checkpoint inhibitor; TKI: tyrosine kinase inhibitor; Ab: antibody.

**Table 2 cancers-13-03666-t002:** Association between alterations in oncogenic signaling pathways and the degree of CD8^+^ tumor infiltrating lymphocytes.

Oncogenic Pathway	Mutation	CD8^+^ TILs		*p* Value ^2^
		Median^1^	25–75th Percentile	
Wnt/β-catenin path.	with	6.18	1.30–26.9	0.0082
	without	17.6	5.77–38.0	
p53/cell cycle path.	with	18.7	5.70–32.7	0.7505
	without	12.9	3.60–38.3	
PI3K/Akt path.	with	1.14	0.17–2.03	0.5836
	without	1.16	0.36–2.88	
Chromatin remodeling	with	17.3	2.16–31.1	0.8056
	without	14.1	4.44–36.0	
Epigenetic regulator	with	0.75	0.15–1.81	0.1488
	without	1.29	0.42–2.88	
Oxidative/ER stress	with	1.63	0.53–5.74	0.1871
	without	1.11	0.28–2.72	
DNA repair	with	1.24	0.45–2.97	0.7392
	without	1.14	0.28–2.73	
TERT promoter	with	1.40	0.38–2.81	0.5093
	without	1.03	0.28–2.70	

Degree of tumor infiltrating lymphocytes (TILs) are compared between HCCs with mutations in each oncogenic pathway and those without mutations. In total, 154 HCCs were examined for mutations using the Ion AmpliSeq Comprehensive Cancer Panel, and the degree of CD8^+^ cells was examined using immunohistochemistry. ^1^ Median: median number of CD8+ TILs/high power field. ^2^
*p* value by Wilcoxon’s rank-sum test.

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
