# Peer review of "Role of Oncogenic Pathways on the Cancer Immunosuppressive Microenvironment and Its Clinical Implications in Hepatocellular Carcinoma"

_cancers, 2021, doi:10.3390/cancers13153666_

Round 1

Reviewer 1 Report

In this review, author provides comprehensive discussion about the cellular component, signaling pathways and tumor microenvironment within HCC tumors, and also impact of DNA mutation on responses of anti-PD-L1 therapy in patients with HCC. I think that either the basic scientific researchers or clinicians will be interesting to read this review. Especially, it's important to explore the potential useful predictors for patients received the ICIs alone or combination with MTAs. I only have minor points for this review:

(1). Reference 59: the dot number has been available, please update (doi.org/10.1159/000516899)

(2). Author has mentioned about the mutation of Wnt/beta-catenin pathway (CTNNB1, AXIN1, and APC) in decreasing expression of IFN-gamma. In addition, author also mentioned about role of mutation of ARID2 in immunosuppressive tumor microenvironment of HCC. Both of them might contribute for responses of anti-PD-L1 in patients with HCC. So, how about anti-PD-L1 responses in patients with mutation of ARID2 and Wnt/beta-catenin in tumors? 

(3). How about the role of mutant Wnt/beta-catentin in patients' response to ICI combination with MTAs (page 8-9, table 1)? Is it still important? I think the readers should be interesting to know that.

Author Response

Response to the Reviewers comments:

Reviewer #1

In this review, author provides comprehensive discussion about the cellular component, signaling pathways and tumor microenvironment within HCC tumors, and also impact of DNA mutation on responses of anti-PD-L1 therapy in patients with HCC. I think that either the basic scientific researchers or clinicians will be interesting to read this review. Especially, it's important to explore the potential useful predictors for patients received the ICIs alone or combination with MTAs. I only have minor points for this review:

Response

Thank you for the valuable suggestions regarding the construction of the review. I have learned the comments carefully and corrected the manuscript. I’m grateful for the positive comments for this review article.

Inquiry 1

Reference 59: the doi number has been available, please update (doi.org/10.1159/000516899).

Response to inquiry 1

Thank you for the comment. I updated the doi in ref. 69.

Inquiry 2

Author has mentioned about the mutation of Wnt/beta-catenin pathway (CTNNB1, AXIN1, and APC) in decreasing expression of IFN-gamma. In addition, author also mentioned about role of mutation of ARID2 in immunosuppressive tumor microenvironment of HCC. Both of them might contribute for responses of anti-PD-L1 in patients with HCC. So, how about anti-PD-L1 responses in patients with mutation of ARID2 and Wnt/beta-catenin in tumors?

Response to inquiry 2

Thank you for the valuable suggestions.

Previously, I have reported that PD-L1 expression in HCC is less common in the tumor with the activating mutations in the Wnt/beta-catenin pathway. However, there are no association between mutations in chromatin remodeling pathway, including ARID2, and degree of CD8+ TILs as well as PD-L1 expression (ref. 33). As mutation of ARID2 is, reportedly, associated with TAM subclass of HCC, immune suppressive mechanism in HCC with ARID2 mutation should be different form that of CTNNB1 (ref. 14). We stated this discussion in page 12, line 435 - 447.

Inquiry 3

How about the role of mutant Wnt/beta-catentin in patients' response to ICI combination with MTAs (page 8-9, table 1)? Is it still important? I think the readers should be interesting to know that.

Response to inquiry 3

I totally agree the efficacy of combination therapy in HCC with immune cold phenotype quite important. Unfortunately, this issue has not been clarified yet. Currently, combination of atezolizumab + bevacizumab is applicable for unresectable HCC; we are now investigating the efficacy of this combination on HCC showing immune cold phenotype. We discussed this point in page 12, line 468 – page 13, line 477.

Reviewer 2 Report

This review is well written, quite complete and effective in reporting the cellular and molecular scenario that makes the HCC microenvironment strongly immunosuppressive.

However, it lacks some topics that could greatly improve it.

  1. Authors should discuss more in depth why immune checkpoint blockade (i.e., anti-PD-1 or anti-PD-L1 alone) is effective only in a minority of virus-induced HCC and almost not at all in HCC with NASH and consider point-by-point the following possibilities reported in various papers: low PD-L1 expression in liver? Overwhelmed tolerance mechanisms in liver? The possibility that the HBV/HCV- or NASH-related diseases preceding HCC and having an average life significantly much longer than other tumors, can condition the development of fully-exhausted T cells that are not susceptible of rescue by ICB? Expansion of PD-1+ Tregs by anti-PD-1 therapy??
  2. In this context, authors should emphasize the role of PD-1 expression on Tregs addressed to inhibit excessive Treg-mediated suppression in chronic HCV infection or cirrhosis representing a pre-TME (Franceschini et al JCI 2009) and discuss that the ICB can be detrimental for this reason in HCC.
  3. At the level of paragraph on Wnt/b-catenin pathway, authors should discuss papers reporting the role of Wnt/b-catenin signaling in inducing T cell dysfunction in HCC and CRC and that β-cat+ CD8 T cells can be rescued by Wnt blockade, resulting in the tumor shrinkage in mouse models (Schinzari V et al, Cancer Immunol Res 2018; Pacella I, Cancer Immunol Res 2018).

Author Response

Response to the Reviewers comments:

Reviewer #2

This review is well written, quite complete and effective in reporting the cellular and molecular scenario that makes the HCC microenvironment strongly immunosuppressive. However, it lacks some topics that could greatly improve it.

Response

Thank you for the kind suggestions and your interests for the review. I have learned the comments carefully and revised the manuscript as follows.

Inquiry 1

Authors should discuss more in depth why immune checkpoint blockade (i.e., anti-PD-1 or anti-PD-L1 alone) is effective only in a minority of virus-induced HCC and almost not at all in HCC with NASH and consider point-by-point the following possibilities reported in various papers: low PD-L1 expression in liver? Overwhelmed tolerance mechanisms in liver? The possibility that the HBV/HCV- or NASH-related diseases preceding HCC and having an average life significantly much longer than other tumors, can condition the development of fully-exhausted T cells that are not susceptible of rescue by ICB? Expansion of PD-1+ Tregs by anti-PD-1 therapy??

Response to inquiry 1

Thank you for the important suggestion. I agree this comment is quite important for the improvement of the quality of the manuscript.

As this is a critical issue, I have added a new head as “3. Unique aspects of immunological characteristics in liver and hepatocarcinogenesis” for this discussion in page 5, line 197 – page 6, line 233. And issues regarding the efficacy of ICB in HCC cases are summarized there, including mutation burden, overwhelmed tolerance mechanism in liver, development of T-cell exhaustion in chronic hepatitis, and expansion of PD-1-positive Tregs by anti-PD-1 therapy. Some critical issues for the immunity of NASH-related HCC, and hyperprogression of tumor on anti-PD-1 therapy are also included here. I also added new references (refs. 36 – 45) for the discussions.

Inquiry 2

In this context, authors should emphasize the role of PD-1 expression on Tregs addressed to inhibit excessive Treg-mediated suppression in chronic HCV infection or cirrhosis representing a pre-TME (Franceschini et al JCI 2009) and discuss that the ICB can be detrimental for this reason in HCC.

Response to inquiry 2

Thank you for the suggestion. I have included this discussion in the new head of “3. Unique aspects of immunological characteristics in liver and hepatocarcinogenesis” in page 5, line 225 – page 6, line 233. We have also discussed the related issue, hyperprogression of tumor on anti-PD-1 antibody, where ICB can be quite detrimental in such cases. I also added new references (refs. 43 - 45) for the discussions.

Inquiry 3

At the level of paragraph on Wnt/b-catenin pathway, authors should discuss papers reporting the role of Wnt/b-catenin signaling in inducing T cell dysfunction in HCC and CRC and that β-cat+ CD8 T cells can be rescued by Wnt blockade, resulting in the tumor shrinkage in mouse models (Schinzari V et al, Cancer Immunol Res 2018; Pacella I, Cancer Immunol Res 2018).

Response to inquiry 3

Thank you for your important suggestion. I have added the discussion regarding this matter in page 7, line 318 – page 8, line 323, and added the new reference in refs. 41, 42.
